# Youth engagement and social innovation in health in low-and-middle-income countries: Analysis of a global youth crowdsourcing open call

Rayner Kay Jin Tan [1,2,3☯] *, Wenjie Shan [1,4☯], Eleanor Hummel[1,2☯], Joseph Deji [5], Yusuf Babatunde [6], Ronald Mirondo Waiswa[7], Ying Zhang [8], Yusha Tao[1], Weiming Tang[1,2], Meredith del Pilar-Labarda [9], Beatrice Halpaap[10], Joseph D. Tucker[2,11]

1 University of North Carolina Project-China, Guangzhou, China, 2 Institute for Global Health and Infectious Disease, University of North Carolina at Chapel Hill, Chapel Hill, North Carolina, United States of America, 3 Saw Swee Hock School of Public Health, National University of Singapore and National University Health System, Singapore, Singapore, 4 Department of International Clinic, Shanghai Children's Medical Center, Shanghai Jiao Tong University School of Medicine, Shanghai, China, 5 Pioneer Medical Initiative, Akure, Nigeria, 6 Faculty of Pharmaceutical Sciences, University of Ilorin, Ilorin, Nigeria, 7 Village Based Rehabilitation Program, Uganda, 8 Faculty of Medicine, School of Translational Medicine, Nursing and Health Sciences, Monash University, Melbourne, Australia, 9 Department of Medicine, University of the Philippines Manila—School of Health Sciences, Palo, Leyte, Philippines, 10 UNICEF/UNDP/World Bank/ WHO Special Programme for Research and Training in Tropical Diseases (TDR), Geneva, Switzerland, 11 Clinical Research Department, London School of Hygiene and Tropical Medicine, London, United Kingdom

☯ These authors contributed equally to this work.
* Rayner.tan@nus.edu.sg

**Data Availability Statement:** Anonymized data are within the paper and its Supporting Information files.

## Abstract

Social innovation in health is a ground-up, community-engaged process that draws on the diverse strengths of local individuals to drive social change and health improvement. Social innovation may be particularly useful in low and middle-income countries to ensure effective and sustainable health solutions. The purpose of this study is to describe the findings of a global youth (18–35 years old) crowdsourcing open call on social innovations, and to identify the levels of engagement in such innovations. We organized a global crowdsourcing open call (Go Youth!) to identify and recognize youth social innovations in health and adopted both quantitative and qualitative approaches to analyze our data. For quantitative analyses, we described the socio-demographic characteristics of youth who submitted innovations. For qualitative analyses, we adopted a deductive-inductive analytic approach utilizing an adapted Hart's Ladder as a conceptual framework for our thematic analysis of participants' submissions, which comprised four levels of youth engagement: none, minimal, moderate, and substantial. The open call received 99 eligible submissions. Most participants were 23 years of age or older (90.7%), resided in LMICs (98.0%), male (64.3%), and had a bachelor's or higher degree (72.4%). Most of the submissions were written in English (93.9%), located in Africa (69.7%), and had prior implementation (60.2%). A total of 39 innovations had substantial youth engagement and qualitative data suggested that youth leadership and peer mentorship of other youth in the community were important aspects of engagement.

**Funding:** The funders had no role in study design, data collection and analysis, decision to publish, or preparation of the manuscript.

**Competing interests:** The authors have declared that no competing interests exist.

LMIC youth developed and implemented social innovations that had evidence of impact or effectiveness in their communities, illustrating how social innovation approaches may be feasible in LMICs. More efforts should be made to identify and empower youth in these settings to spark change.

## Introduction

Social innovation in health is a bottom-up, community-engaged process that draws on diverse strengths of local individuals to link social change and health improvement [1]. There is growing evidence that social innovation in health approaches can improve health outcomes [2,3] and address the upstream determinants of health in low and middle income countries (LMICs) [4,5]. Examples include the use of crowdsourcing to develop effective community-driven, bottom-up solutions, or the use of crowdfunding and public engagement platforms in LMIC settings to fund community-driven health research [6,7]. These promising findings have also paved the way for the assimilation of social innovation in medical education [8], implementation science [9], community engagement [10], and digital health [11].

The Social Innovation in Health Initiative (SIHI) was launched with the support of UNICEF/UNDP/World Bank/WHO Special Programme for Research and Training in Tropical Diseases (TDR) to champion research, capacity-building, and advocacy in LMICs.[1] Social innovations in health are often low-cost and tailored for users in resource-limited settings [12]. The ground-up orientation of social innovations in health may also be particularly suited to recognize endogenous forms of innovations in LMICs and disrupt entrenched systems to advance the sustainable development goals.[1] One way to identify social innovations is through crowdsourcing.

Crowdsourcing has non-experts and experts generate solutions to a problem, then share selected solutions with the broader public [13]. This approach has been used to improve health outcomes and tap collective wisdom [14,15], engage communities in diverse settings, and identify high-quality innovations [13]. Crowdsourcing open calls have been used to develop research mentorship strategies [16], inform novel health interventions [17,18], and inform health policy [19].

Crowdsourcing can also solicit innovations from youth [20,21]. While crowdsourcing studies have focused on youth in single countries or regions, fewer studies have examined youth social innovation in a global context. Furthermore, few crowdsourcing open calls consider the depth of community engagement in their submissions. To address these gaps, we draw on our experience of organizing the Go Youth! global crowdsourcing open call to identify social innovations in health from youth aged 18 to 35 years old. This open call built on previous youth social innovation crowdsourcing calls in Malaysia and the Philippines organized in 2021 by the Social Entrepreneurship to Spur Health team, supported by the Social Innovation in Health Initiative. Evidence from these crowdsourcing open calls suggested that youth were capable of developing high-quality creative health solutions in diverse settings [20,21]. The purpose of this study is to describe youth-led social innovations submitted to a global open call and describe the extent of youth engagement in the innovations.

## Methods

### Objectives of the global open call and organizing a steering committee

We organized our crowdsourcing open call according to the TDR practical guide (Fig 1) [15]. Data were collected from March 7, 2022 to June 30, 2022. We organized a steering committee

responsible for high-level advice and an organizing committee charged with implementation. The 17-person steering committee included representatives from organizations involved in youth mobilization and health promotion. Eight members of the steering committee were youth. Steering committee meetings were held once every two months and involved a one-hour teleconference meeting to provide feedback on the call for submissions, suggest potential judges, and finalize the prize structure. The promotion and engagement components were led by a four-person organizing committee comprising three youth researchers and a mentor in social innovations in health.

## Engaging the community to contribute

The open call accepted submissions over a 12-week period. Participants were encouraged to submit their innovations as long as they fell within one of three topics: health topics, health systems, and messaging. More details can be found in **S1 Table**. Overall, submissions were accepted as text (maximum 500 words), or video/audio (maximum three minutes). Submissions were accepted in the six languages of the United Nations: Arabic, Chinese, English, French, Spanish, and Russian.

Our team partnered with 20 organizations to promote the open call. These organizations typically serve youth in their own contexts and have experience tailoring promotional material to better resonate with their own youth members. A capacity-building webinar for youth to participate and share their ideas for submission was organized and six facilitators from the Social Entrepreneurship to Spur Health team facilitated small group discussions in breakout rooms with 48 youth. The open call website received 3422 unique visitors, while the open call twitter post received 3857 impressions and 195 engagements (inclusive of retweets, link clicks, and likes) during the span of our promotional period. Our team worked with partner organizations to conduct promotional campaigns in several settings, which led to the development of translated promotional material, radio interviews, and radio public service announcements.

As shown in **Fig 1**, three additional capacity building activities were conducted to scaffold our approach to the entire open call process. This is a novel component of our crowdsourcing open call that builds on the existing conceptual framework and stages of crowdsourcing from the WHO practical guide for crowdsourcing in health [15]. We adopted a scaffolding approach, where multiple stages of learning and key milestones are incorporated to provide the necessary assistance to enable participants and learners to accomplish tasks and develop an understanding of the learning material [22].

## Receiving and evaluating contributions

A judging panel was convened to assess each submission and was comprised of health professionals, researchers, youth community leaders and social innovators. Judges were selected based on their expertise in working with youth in the community, experience with implementing social innovations in health, as well as health systems researchers and professionals who have experience in research on social innovations in health. Seventeen independent judges used pre-defined judging criteria which were developed by the steering committee to evaluate submissions, including (i) clarity and conciseness, (ii) relevance, (iii) novelty, (iv) feasibility, scalability/replicability and sustainability, and (v) promotion of equity and fairness (**S2 Table**). Each judge was briefed on the judging criteria, and confirmed that they had understood the criteria and scoring protocol prior to the judging process. All submissions were randomly assigned to judges for scoring. Non-English submissions were translated to English for judging. Multimedia submissions were transcribed. Judges were asked to provide a single score of 1 to 10 and were asked to recuse themselves from submissions where a conflict of interest was

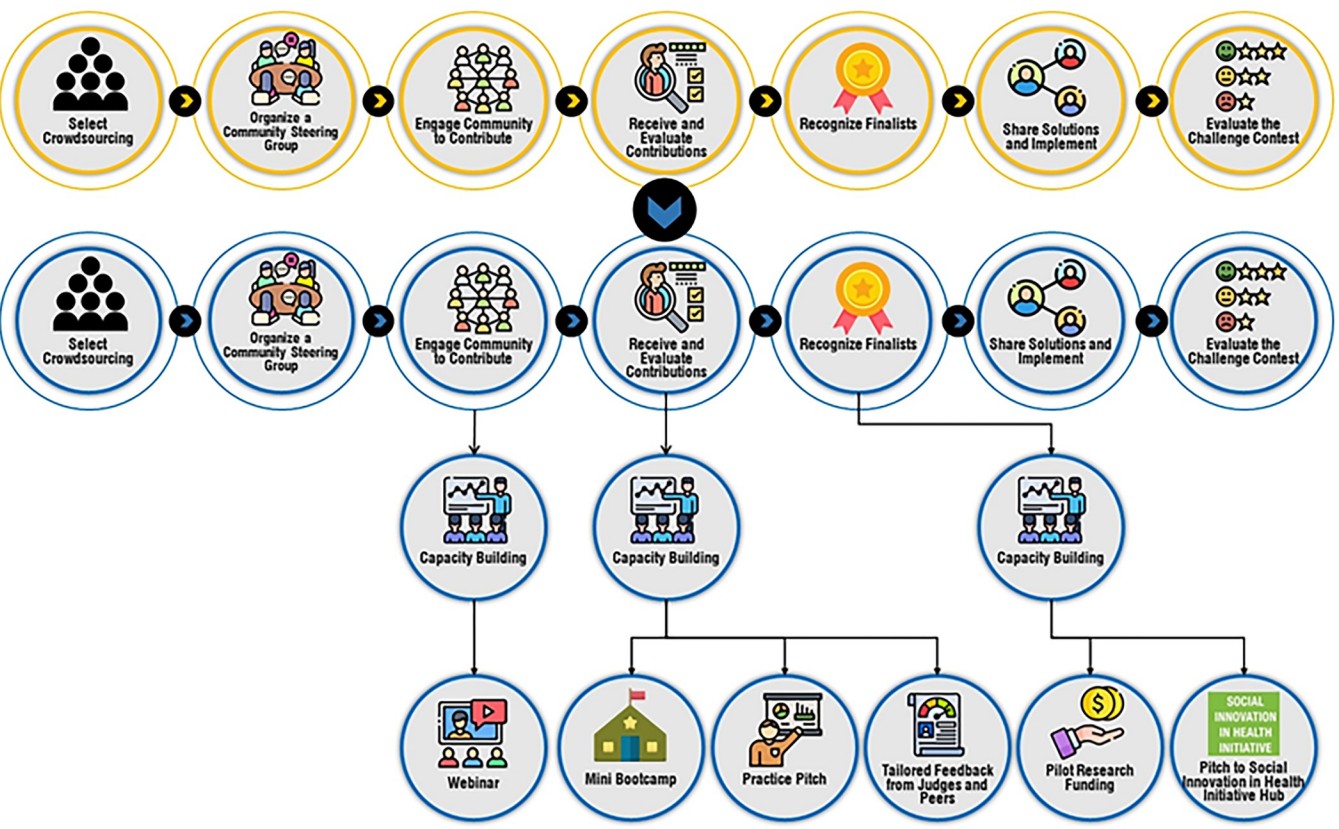

**Fig 1. Expansion of capacity-building activities in conventional crowdsourcing stages.**

identified. Each submission received at least three scores, and additional judges were assigned to submissions where scores exceeded a standard deviation of two. A total of eleven semi-finalists were identified following the judging process (three participants were tied in ninth place, and therefore there was a total of 11 submissions selected for the semi-finals).

## Recognizing finalists and sharing solutions

Youth were recognized and supported at several stages of the open call. Each youth participant was invited to participate in six training workshops to build capacity for social innovations in health. Each youth or youth team received collated feedback about strengths and weaknesses of applications overall and a copy of this manuscript. Semi-finalists, who were identified by having a mean score of 7/10 or greater, were invited to two videoconference capacity building workshops. The first was a practice pitch event where participants delivered a five-minute pitch and received peer and expert feedback. Participants were then asked to fill out a needs assessment survey and this collated information on needs directly informed a mini bootcamp. Based on the feedback, we organized a third capacity-building activity, a crowdfunding bootcamp, to help teams better prepare for their public pitch as well as to help them crowdfund their social innovations in the future. These activities culminated in a public pitch, where all 11 finalists presented their innovations through 5-minute video pitches, which were evaluated by a panel of four independent judges. A total of three finalists were selected to receive between USD350 to USD950 research seed funding for their ideas.

### Quantitative data analysis

We conducted descriptive analysis of submissions received through the open call. Submission characteristics, such as the country of implementation, mean judging score, team size, submission language, and prior implementation, were assessed. At the individual level, demographic attributes were assessed.

### Qualitative data generation and analysis

We conducted a qualitative data analysis to identify and understand the depth of youth engagement in the open call submissions. While all of the description of innovations were submitted by youth, this analysis was especially focused on how these youth-designed innovations engaged other youth in their communities. To do so, we analyzed two sources of data within the submissions; first, we analyzed responses to a question that asked participants about how youth were engaged in the submission, and second, we analyzed the innovation descriptions themselves. Qualitative data was derived from participants' submissions. Text-based submissions were used directly for analysis, while data from videos and audio sources were transcribed verbatim, and if necessary, subsequently translated in English prior to analysis. Coding was conducted by two qualitative analysts (RT and EH). We adopted a deductive-inductive analytic approach utilizing an adapted Hart's Ladder as a conceptual framework for analysis (Fig 2). Hart's Ladder differentiates youth engagement into four levels–substantial, moderate, minimal, or none [23].

Substantial engagement is the highest level of youth engagement. Youth co-lead the research team and receive training and mentorship to drive the development of the innovation. Moderate engagement involves shared leadership with youth. While youth are not directly designing or creating the innovation, they are an active part of the team and are tasked with managing important projects.

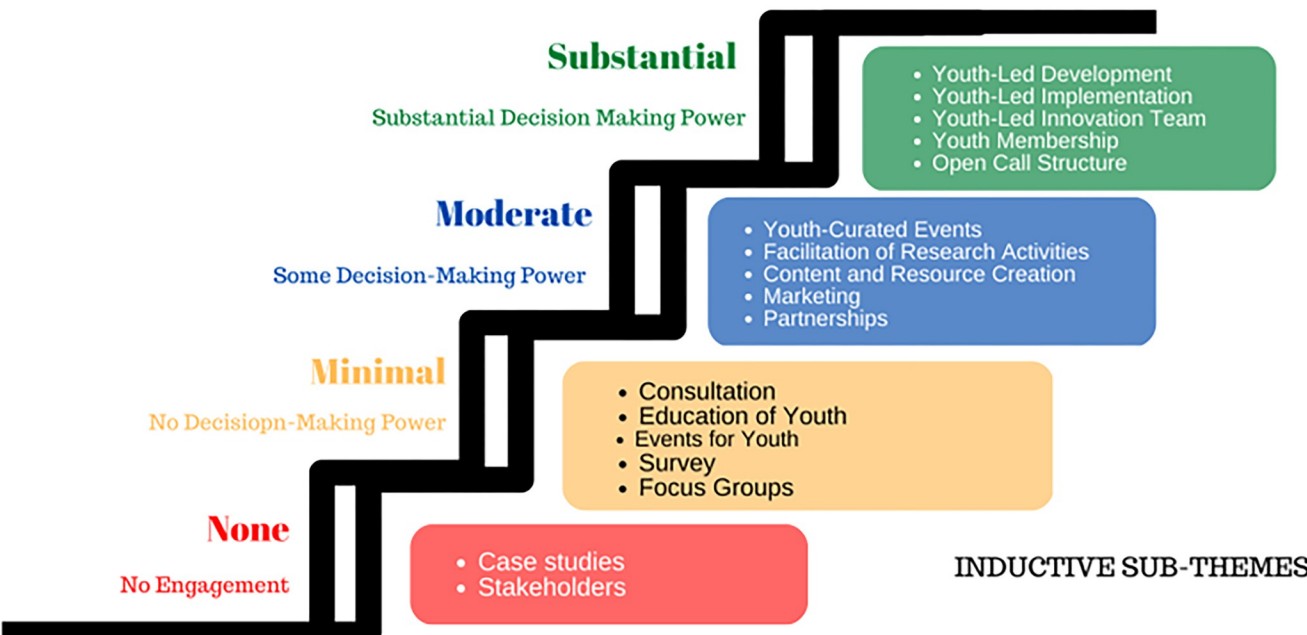

**Fig 2. Adapted Hart's ladder categorizing youth engagement.**

Minimal engagement incorporates young people's opinion, yet the power to implement changes and make decisions about the project is not shared. While stakeholders may raise concerns, there is no guarantee that they will receive an answer from the project leads. Finally, non-engagement may consider youth as stakeholders or consult prior youth-led research. No youth are engaged to be a part of any aspect of the innovation.

If a submission indicated multiple forms of youth engagement, the highest level of engagement was assigned to that submission for the comparison of scores. Following deductive grouping of our codes, inductive coding proceeded to generate sub-categories illustrating specific forms of youth engagement (e.g., youth-led development). We then sought to descriptively analyse differences in scores across submissions that engaged in different levels of youth engagement.

### Patient and public involvement

Key stakeholders were involved in the planning stages of the open call as steering committee members. Youth were included on the organizing committee, steering committee, and among our open call participants. In turn, the steering committee provided feedback on appropriate submission pathways (e.g., inclusion of non-text-based submissions to encourage submissions from youth who may prefer to talk about their innovations), evaluations (e.g., judging criteria that would be fair and suitable for youth), commendation (e.g., prizes and recognition that would be most useful for youth), and overall design of the open call. Finally, we invited youth finalists (JD, YB, RW) from our open call to co-develop and contribute to this paper.

### Ethics statement

This study was determined to be IRB-Exempt by the University of North Carolina at Chapel-Hill IRB (Ref: 23–2285). Formal written consent was obtained from participants.

## Results

### Participant and submissions characteristics

The open call received a total of 99 entries, among which 98 participants provided complete socio-demographic information. The characteristics of the submissions are summarized in **Table 1**, and the overview of the submissions is presented in **Table 2**. Most of these participants were 23 years of age or older (90.7%), male (64.3%), had a bachelor's or higher degree (72.4%), identified as being straight or heterosexual (80.6%), and resided in LMICs (98.0%). Most of the submissions were written in English (93.9%), located in Africa (69.7%; also see **Fig 1**), and had prior implementation (60.2%). An overview of the themes of the entries is presented in **S3 Table**.

The average score of all the entries was 5.8±1.6 and the distribution of the mean scores is illustrated in **S1 Fig**. We described the characteristics of submissions with different score brackets and found that the submissions with data on prior implementation had higher scores (**S4 Table**, p = 0.043).

### Youth engagement in the open call

We identified substantial (n = 39), moderate (n = 12), minimal (n = 18), and no (n = 30) youth engagement in these social innovations. **Fig 2** summarizes these concepts around youth engagement, while **Table 3** summarizes the themes and illustrative quotes from participants.

Substantial engagement had five sub-themes: youth-led innovation teams (35%), youth-led development (31%), youth-led implementation (29%), youth mentorship (4%), and youth

**Table 1. Sociodemographic characteristics of Global "Go Youth" open call participants, 2021–2022 (n = 99).**

| Variable | | Count, n (%) |
|---|---|---|
| Age (years) | 18–22 | 9 (9.3) |
| | 23–26 | 28 (28.9) |
| | 27–30 | 30 (30.9) |
| | 31–35 | 30 (30.9) |
| Gender | Male | 63 (64.3) |
| | Female | 34 (34.7) |
| | Transgender | 1 (1.02) |
| Highest degree | High school graduate, diploma, or the equivalent | 23 (23.5) |
| | Bachelor's degree | 38 (38.8) |
| | Masters or similar professional degree | 26 (26.5) |
| | Doctoral degree | 7 (7.1) |
| | Others | 4 (4.1) |
| Sexuality | Straight | 79 (80.6) |
| | Bisexual | 5 (5.1) |
| | Gay or lesbian | 3 (3.1) |
| | Not sure | 7 (7.1) |
| | Refuse to answer | 3 (3.1) |
| | Another sexual orientation | 1 (1.0) |
| Residence | LMIC | 96 (98.0) |
| | HIC | 2 (2.0) |

Abbreviation: LMIC, low and middle income countries; HIC, high income countries.

open calls (2%). Oftentimes, the same response would mention multiple forms of substantial engagement, indicating that these avenues to involve youth often were used in tandem with each other. Codes of innovations that involved youth in every aspect of the process often emphasized youth leadership and consistent involvement:

> *"The design, implementation, and evaluation of this innovation have primarily been led by youth. Second, a core group of about ten youth spearheaded both the program implementation and data collection activities for the innovation's pilot."*

Another important part of substantial youth engagement that emerged was mentorship of youth and by youth to develop research skills.

Moderate engagement consisted of outreach (13%), marketing (9%), content creation (13%), research activities (52%), youth-led events (9%), and partnerships (4%). These sub-themes focused on youth leadership in smaller projects or tasks related to the innovation. For example, one submission described youth engagement in data collection and outreach.

> *"We have 12 volunteers who are youths and students in Ibadan, Nigeria, who participate in outreaches to Primary Health Care and will assist in collecting data to evaluate the impact of our innovation."*

Involving youth with research activities like this one was the most frequently used form of moderate youth involvement. Coding revealed that youth were also often asked to design marketing materials and create content for innovations, such as educational tools.

Minimal engagement included focus group discussions (7%), surveys (11%), youth events (7%), educational programming (11%), and consultation (59%). These submissions typically

**Table 2. Characteristics of all the submissions in the global "Go Youth" open call, 2021–2022 (n = 99).**

| Variable | Count, n (%) |
|---|---|
| **Region where the innovation is being implemented** | |
| Africa | 69(69.7) |
| Asia and the Pacific | 17(17.2) |
| Latin America and the Caribbean | 6(6.1) |
| Europe | 1(1.0) |
| North America | 1(1.0) |
| Multiple countries | 5(5.1) |
| **The number of team members** | |
| One | 53(53.5) |
| Two | 16(16.2) |
| Three | 17(17.2) |
| Four | 13(13.1) |
| **Language of submission** | |
| English | 93(93.9) |
| Spanish | 4(4.0) |
| French | 1(1.0) |
| Arabic | 1(1.0) |
| **Ever participated in social innovation in health activities before** | |
| No | 44(46.3) |
| Yes | 51(53.7) |
| **Prior implementation on innovation** | |
| No | 39(39.8) |
| Yes | 59(60.2) |
| **Mean scores (SD)** | 5.8(1.6) |

SD: Standard deviation.

focused on soliciting opinions from youth. Some forms of minimal engagement are more intensive than others. For instance, one innovation consulted a focus group of youth and asked for their opinions on important aspects of design and implementation.

> *"We usually have a small group meeting when we want to make a decision, these are some carefully selected youth in the society that we fell like can represent the entire society."*

While this is minimal engagement, this approach was bi-directional and long-term as youth were consistently consulted throughout the process. Less intensive forms of minimal engagement were uni-directional or asked about narrow aspects of an innovation.

There were two major sub-themes of non-engagement: case studies and stakeholders. These innovations considered youth yet lacked active engagement. For instance, one response stated: "Youth mental health experiences. . .served as a basis for my motivation to write this concept." In most cases of non-engagement, youth are stakeholders.

## Youth engagement and open call scores

After thematic analysis, the codes were matched with the original submissions. The scores of these submissions and youth engagement codes were then compared (**S5 Table**). The average score for all submissions was 5.8. Submissions that did not describe any youth engagement had the lowest average score of 5.3. The average score for submissions that had minimal youth

**Table 3. Themes for youth engagement.**

| Theme | Sub-Theme | Illustrative Quote(s) |
|---|---|---|
| No Engagement | Case Studies | "Youths in technological advanced worlds like the United States have designed, implemented excellent transportation inventions leading to massive growth in economy. An example is Bolt created by Markus Villig, 28 years."<br>"The GoFundMe project was designed, implemented, and evaluated by youths Brad Damphousse and Andrew Ballester 12 years ago." |
| | Stakeholders | "Youth provided the inspiration behind this innovation. Youth mental health experiences with mental health and their coping mechanisms served as a basis for my motivation to write this concept." |
| Minimal Engagement | Consultation | "While shaping the idea for this innovation, I also sought feedback from youth within my network. As a group, they represented different backgrounds and cultures that reflect the initial target demographic."<br>"We have also consulted with young medical professionals and youth in our pilot community in the creation of our product design." |
| | Education of Youth | "I encourage use of school going children from different communities through directly interacting with them, we share practice knowledge and skills with them. These pass information to different homes, I also encourage direct participation of those I find in communities including the parents as many parents of clients with disabilities are youths." |
| | Events for Youth | "We also would like to conduct the camping to connect between youth and youth who are vision impairment as we want them to share their experiences and do activities together by non-discrimination." |
| | Surveys | "Different forms were passed asking youth various questions and thoughts on the idea of expecting mothers planning early for their future babies and feedback was received." |
| | Focus Groups | "We usually have a small group meeting when we want to make a decision, these are some carefully selected youth in the society that we fell like can represent the entire society."<br>"First, in the early stages, multiple youth carried out focus group discussions that our team conducted to better define the ideas behind the innovation." |
| Moderate Engagement | Youth-Curated Events | "The youth took part of discussions and play making processes to address issues affecting their communities and then also finding amiable solutions to the raised problems using drama." |
| | Facilitation of Research Activities | "During the distribution of the book into the schools, we also sent out invitations to a number of students from Makerere University and Mbarara University of Science and Technology, to be involved in the "read aloud sessions"."<br>"We have 12 volunteers who are youths and students in Ibadan, Nigeria, who participate in outreaches to Primary Health Care and will assist in collecting data to evaluate the impact of our innovation." |
| | Content and Resource Creation | "We designed resources with learners in school aged 14–19 years, teachers between 18 to 34 years, other stakeholders at the ministries of education and sports, health and Gender." |
| | Outreach | "We started engagement from the district level till national level and therefore after several attempts with now luck to decide it was now the time to create campaigns that speaks of young people issue's within the broken healthcare system." |
| | Marketing | "The young people also helped in co-designing the flyers which were distributed to the communities. We had young people involved in the hackathon and helping to co-design the logos and songs which we needed for the branding of our study." |
| | Partnerships | "Partnerships were made with youth led and youth-centric organizations" |
| Substantial Engagement | Youth-Led Development | "Young deaf peer educators were directly involved in designing and developing the solution."<br>"CBITS has been designed and implemented among students from 5th grade through 12th grade who have witnessed or experienced trauma and adversity in their life events such as community and school violence, accidents and injuries, physical abuse and domestic violence, and natural and man-made disasters." |
| | Youth-Led Implementation | "The design, implementation, and evaluation of this innovation have primarily been led by youth. Second, a core group of about ten youth spearheaded both the program implementation and data collection activities for the innovation's pilot." |
| | Youth-Led Innovation Team | "Youths (aged 18 to 35) were involved in this social innovation. From Designing to implementing to scaling, to programming. Every aspect. The Team comprises youths between the ages of 18 to 35." |
| | Youth Mentorship | "We engage youth graduates as researchers and equip them with the requisite skills and capacity" |
| | Open Call Structure | "Open call was made to all youth health professionals to join the innovation. Five(5) youth between the ages of 18 to 35 were involved in the planning stage. During the implementation stage, Seven(7) additional youth were added to the team and have since been working and evaluating the progress of the innovation on quarterly bases." |

engagement was 5.8, while those with moderate level of engagement scored 6.2 on average. Submissions with a substantial level of youth engagement had a score of 6.1.

## Conclusion

Our crowdsourcing open call identified youth-led social innovations from many LMICs. Our results suggest that youth can play leadership roles in social innovations in health. These findings are consistent with other crowdsourcing open calls [20,21]. Our findings contribute to the literature on crowdsourcing open calls, social innovations in health, and youth engagement in LMICs.

We found that youth can lead social innovations in health. This is consistent with data from social innovation open calls in Malaysia and the Philippines [20,21]. The focus of this global open call on innovations that already have some form of pilot implementation or data has not been a focus of past work. This shows that not only do youth have great ideas, but they also have the ability to create solutions in their own communities that are potentially ready to be formally evaluated or scaled up. This study also provides further evidence for the feasibility of social innovations to spur health in LMICs, where top-down funding and resourcing for the strengthening of health systems may be suboptimal. The crowdsourcing open call has identified social innovations that are bottom-up and driven by youth in LMICs and have preliminary data supporting the effectiveness of their innovations. Such efforts can be further cultivated to drive innovation and transform healthcare delivery systems at the local level [1].

Our data suggest that youth-led social innovations can substantially engage youth in LMICs. In this open call, participants' submissions reflected varying depths of youth engagement. In our analyses of scores, participants and projects that engaged in varying forms of engagement obtained the highest scores. These findings corroborate the findings of studies analysing youth engagement in other crowdsourcing open calls, which also found that youth were involved across projects in similarly varying levels or depths of engagement [24]. These also align with Hart's ladder of youth engagement. In a scoping review on youth engagement in HIV prevention intervention research in sub-Saharan Africa, youth engagement was also categorized through an adapted Hart's ladder, including levels of 'substantial', 'moderate', 'minimal', and 'no' youth engagement [23]. While results were not statistically significant, our study provides descriptive evidence that submissions and innovations that go beyond just representation and meaningfully engage youth in the development and implementation of such innovations, achieved higher scores based on the judging criteria of the open call.

Finally, our open call included more capacity building activities compared to other open calls [14,15]. This was especially important because about half of the youth reported participating in social innovation in health activities prior to the open call. In the context of the crowdsourcing open call, the first participatory webinar aimed to clarify concepts around the judging criteria and provide participants a space to clarify doubts with their submissions. The practice pitch then provided semi-finalists the opportunity to prepare for their final pitch in terms of the format and process, as well as get structured feedback from mentors and other peers. The mini bootcamp focused on crowdfunding components that allowed participants to consider how their respective pitches could incorporate elements of crowdfunding. Finally, seed funding was given to the top three submissions to spur further research and scaling up. Further research on how such capacity-building activities better improve outcomes for crowdsourcing open calls and build capacity for youth in LMIC settings is warranted.

This study has several limitations. First, while we had successfully mitigated risks of low participation or engagement in this open call [15], and added on innovative approaches to engage youth participants through additional capacity-building activities, more formal evaluation work should be conducted to better understand the capacity-building activities. Second,

while our qualitative analysis found that several teams had little to no youth engagement beyond the submitting team's efforts, or that these were not sufficiently described by the team. Clearer instructions and further process evaluation methods to qualitatively explore the nature of youth engagement among participants is warranted. Third, we note that there were certain demographic imbalances in those who submitted to our open call, including greater representation from those who were of the male gender, and those with at least a bachelor's degree and above. These reflected demographics of the submitting participant, but we did not collect information from other team members. Future calls should consider collecting more detailed information from participants to better elucidate potential inequities in the submissions process.

Nevertheless, strengths of this study included using both quantitative and qualitative approaches to describe a more significant whole. As social innovations in health is an emerging field that has shown promise in sparking local action to address local needs, our study can guide further efforts for crowdsourcing, as well as social innovation approaches that aim to engage youth. The strong input of LMIC partner organizations helped to galvanize momentum and may have facilitated the process.

In conclusion, future research could be done in multiple areas. First, research should focus on how capacity building activities in crowdsourcing can be evaluated to support such social innovation activities and build capacity for LMIC social innovators; second, further work could be done to compare how youth engagement or social innovations may differ from one setting or context to others; third, more work can be done in the monitoring and evaluation of such innovations identified by crowdsourcing open calls and to support the sustainable implementation of such innovations in the long run.

## Supporting information

**S1 Checklist. Inclusivity in global research.**
(DOCX)

**S1 Fig. Histogram showing a distribution of mean scores of entries to the global "Go Youth" open call, 2021–2022 (n = 99).**
(DOCX)

**S1 Table. Submission themes for the Go Youth! global open call.**
(DOCX)

**S2 Table. Judging Criteria for the Go Youth! global open call.**
(DOCX)

**S3 Table. Overview of the themes of the entries to the global "Go Youth" open call, 2021–2022 (n = 99).**
(DOCX)

**S4 Table. Descriptive statistics for each variable stratified by scores of submissions in the global "Go Youth" open call, 2021–2022 (n = 99).**
(DOCX)

**S5 Table. Scores for submissions based on types of youth engagement.**
(DOCX)

## Acknowledgments

We would like to thank all participants in the open call. The work received support from TDR, the Special Programme for Research and Training in Tropical Diseases co-sponsored by

UNICEF, UNDP, the World Bank and WHO. TDR is able to conduct its work thanks to the commitment and support from a variety of funders. These include our long-term core contributors from national governments and international institutions, as well as designated funding for specific projects within our current priorities. For the full list of TDR donors, please visit TDR's website at: https://www.who.int/tdr/about/funding/en/ TDR receives additional funding from Sida, the Swedish International Development Cooperation Agency, to support SIHI.

## Author Contributions

**Conceptualization:** Rayner Kay Jin Tan, Eleanor Hummel, Ying Zhang, Yusha Tao, Weiming Tang, Meredith del Pilar-Labarda, Beatrice Halpaap, Joseph D. Tucker.

**Data curation:** Rayner Kay Jin Tan, Wenjie Shan, Eleanor Hummel, Ying Zhang, Yusha Tao, Weiming Tang, Joseph D. Tucker.

**Formal analysis:** Rayner Kay Jin Tan, Wenjie Shan, Eleanor Hummel, Joseph Deji, Yusuf Babatunde, Ronald Mirondo Waiswa, Ying Zhang, Yusha Tao, Weiming Tang, Joseph D. Tucker.

**Funding acquisition:** Rayner Kay Jin Tan, Weiming Tang, Meredith del Pilar-Labarda, Beatrice Halpaap, Joseph D. Tucker.

**Investigation:** Rayner Kay Jin Tan, Wenjie Shan, Eleanor Hummel, Joseph Deji, Yusuf Babatunde, Ronald Mirondo Waiswa, Ying Zhang, Yusha Tao, Weiming Tang, Joseph D. Tucker.

**Methodology:** Rayner Kay Jin Tan, Joseph D. Tucker.

**Project administration:** Rayner Kay Jin Tan, Wenjie Shan, Eleanor Hummel, Ying Zhang, Yusha Tao, Weiming Tang, Joseph D. Tucker.

**Resources:** Weiming Tang, Meredith del Pilar-Labarda, Beatrice Halpaap, Joseph D. Tucker.

**Supervision:** Rayner Kay Jin Tan, Weiming Tang, Meredith del Pilar-Labarda, Beatrice Halpaap, Joseph D. Tucker.

**Visualization:** Rayner Kay Jin Tan, Wenjie Shan, Eleanor Hummel, Joseph Deji, Yusuf Babatunde, Ronald Mirondo Waiswa, Ying Zhang, Yusha Tao, Joseph D. Tucker.

**Writing – original draft:** Rayner Kay Jin Tan, Wenjie Shan, Eleanor Hummel, Joseph Deji, Yusuf Babatunde, Ronald Mirondo Waiswa, Ying Zhang, Yusha Tao, Weiming Tang, Meredith del Pilar-Labarda, Beatrice Halpaap, Joseph D. Tucker.

**Writing – review & editing:** Rayner Kay Jin Tan, Wenjie Shan, Eleanor Hummel, Joseph Deji, Yusuf Babatunde, Ronald Mirondo Waiswa, Ying Zhang, Yusha Tao, Weiming Tang, Meredith del Pilar-Labarda, Beatrice Halpaap, Joseph D. Tucker.

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
