## [Decision Letter · Decision Letter 0]

22 Feb 2024

PGPH-D-23-02030

Youth Social Innovation in Health in Low-and-Middle-Income Countries: A Mixed Methods Analysis of a Global Youth Crowdsourcing Open Call

Dear Dr. Tan,

Thank you for submitting your manuscript to PLOS Global Public Health. After careful consideration, we feel that it has merit but does not fully meet PLOS Global Public Health’s publication criteria as it currently stands. Therefore, we invite you to submit a revised version of the manuscript that addresses the points raised during the review process.

Please address the concerns of the reviewers

We look forward to receiving your revised manuscript.

Kind regards,

Ejemai Eboreime, MD, MSc, PhD

Academic Editor

Journal Requirements:

2. Please send a completed 'Competing Interests' statement, including any COIs declared by your co-authors. If you have no competing interests to declare, please state "The authors have declared that no competing interests exist". Otherwise please declare all competing interests beginning with the statement "I have read the journal's policy and the authors of this manuscript have the following competing interests:"

3. Please amend your detailed Financial Disclosure statement. This is published with the article. It must therefore be completed in full sentences and contain the exact wording you wish to be published.

b. If any authors received a salary from any of your funders, please state which authors and which funders.

If you did not receive any funding for this study, please simply state: “The authors received no specific funding for this work.

4. In the online submission form, you indicated that "Data is available from the corresponding author (RT) upon reasonable request". All PLOS journals now require all data underlying the findings described in their manuscript to be freely available to other researchers, either 1. In a public repository, 2. Within the manuscript itself, or 3. Uploaded as supplementary information.

Additional Editor Comments (if provided):

Reviewers' comments:

Reviewer's Responses to Questions

**Comments to the Author**

1. Does this manuscript meet PLOS Global Public Health’s publication criteria? Is the manuscript technically sound, and do the data support the conclusions? The manuscript must describe methodologically and ethically rigorous research with conclusions that are appropriately drawn based on the data presented.

Reviewer #1: Yes

Reviewer #2: Partly

Reviewer #3: Yes

2. Has the statistical analysis been performed appropriately and rigorously?

Reviewer #1: Yes

Reviewer #2: N/A

Reviewer #3: Yes

3. Have the authors made all data underlying the findings in their manuscript fully available (please refer to the Data Availability Statement at the start of the manuscript PDF file)?

Reviewer #1: Yes

Reviewer #2: No

Reviewer #3: Yes

4. Is the manuscript presented in an intelligible fashion and written in standard English?

Reviewer #1: Yes

Reviewer #2: Yes

Reviewer #3: Yes

5. Review Comments to the Author

Reviewer #1: In the introduction section, regarding “There is growing evidence that social innovation in health approaches can improve health outcomes2 3 and address the upstream determinants of health in low and middle income countries”, please elaborate on how this has been done?

You mentioned three themes in this youth global open call, and I'm curious how you ensured the rigor of the judging? Did each judge randomly review all three categories? How did you ensure that all three categories were evaluated in the fairest way possible? How do you ensure that the judges are consistent in their understanding of the rating scale?

It seems like most of the discussion centered around this contest would increase youth engagement in health topics, but what are its implications for LMIC settings? In other words, what is the significance of this open call, the submissions and the findings? It should not merely complement the existing crowdsourcing contests.

Reviewer #2: This is a very interesting topic and highlights the importance of youth driven innovations in the health sector. The paper will benefit from an improved structural sequence and consistency across all sections as well as more appropriate interpretation of the meaning and significance of findings. Readers will benefit from a better delineation of the different aspects; the study and the intervention (capacity development) including a clear description of whether the impact of the program was assessed as part of the study. Specific details are outlined in the attached document.

Reviewer #3: Dear Authors,

Thank you for sharing this interesting article.

I have several suggestions for improvement:

1. In paragraph 3 of the introduction: The author begins each sentence with "Crowdsourcing". Please consider rewriting this paragraph to enhance readibility

2. Regarding the purpose of the study, one aspect is "to describe the extent of youth engagement in the innovations." In this context, could you clarify who the youth are? Are they the members of the implementing team or the youth affected by the interventions?

Methodology : no comments provided

Results/Discussion

The majority of participants (70%) were from Africa. How do the characteristics of these participants influence the study results? Are there any differences in the type of engagement or innovations observed among non-African participants?

The article mentions that this open call included more capacity building compared to other open calls. Could you explain the rationale behind comparing this call with others? How can you justify that it has a greater capacity-building component?

How do the characteristics of the participants affect the types of innovation observed?

6. PLOS authors have the option to publish the peer review history of their article (what does this mean?). If published, this will include your full peer review and any attached files.

**Do you want your identity to be public for this peer review?** For information about this choice, including consent withdrawal, please see our Privacy Policy.

Reviewer #1: No

Reviewer #2: No

Reviewer #3: No

---

## [Decision Letter · Decision Letter 1]

5 Jun 2024

Youth Engagement and Social Innovation in Health in Low-and-Middle-Income Countries: Analysis of a Global Youth Crowdsourcing Open Call

PGPH-D-23-02030R1

Dear Dr. Tan,

We are pleased to inform you that your manuscript 'Youth Engagement and Social Innovation in Health in Low-and-Middle-Income Countries: Analysis of a Global Youth Crowdsourcing Open Call' has been provisionally accepted for publication in PLOS Global Public Health.

Best regards,

Ejemai Eboreime, MD, MSc, PhD

Academic Editor

Reviewer Comments (if any, and for reference):

Reviewer's Responses to Questions

**Comments to the Author**

1. If the authors have adequately addressed your comments raised in a previous round of review and you feel that this manuscript is now acceptable for publication, you may indicate that here to bypass the “Comments to the Author” section, enter your conflict of interest statement in the “Confidential to Editor” section, and submit your "Accept" recommendation.

Reviewer #1: All comments have been addressed

Reviewer #2: All comments have been addressed

2. Does this manuscript meet PLOS Global Public Health’s publication criteria? Is the manuscript technically sound, and do the data support the conclusions? The manuscript must describe methodologically and ethically rigorous research with conclusions that are appropriately drawn based on the data presented.

Reviewer #1: Yes

Reviewer #2: Yes

3. Has the statistical analysis been performed appropriately and rigorously?

Reviewer #1: Yes

Reviewer #2: N/A

4. Have the authors made all data underlying the findings in their manuscript fully available (please refer to the Data Availability Statement at the start of the manuscript PDF file)?

Reviewer #1: Yes

Reviewer #2: Yes

5. Is the manuscript presented in an intelligible fashion and written in standard English?

Reviewer #1: Yes

Reviewer #2: Yes

6. Review Comments to the Author

Reviewer #1: The authors have addressed my comments and concerns.

Reviewer #2: All comments have been satisfactorily addressed

7. PLOS authors have the option to publish the peer review history of their article (what does this mean?). If published, this will include your full peer review and any attached files.

**Do you want your identity to be public for this peer review?** For information about this choice, including consent withdrawal, please see our Privacy Policy.

Reviewer #1: No

Reviewer #2: No
